# Macrophage-Colony-Stimulating Factor Receptor Enhances Prostate Cancer Cell Growth and Aggressiveness In Vitro and In Vivo and Increases Osteopontin Expression

**DOI:** 10.3390/ijms232416028

**Published:** 2022-12-16

**Authors:** Alexandra Mougel, Eric Adriaenssens, Boris Guyot, Lu Tian, Stéphanie Gobert, Thierry Chassat, Philippe Persoons, David Hannebique, Hélène Bauderlique-Le Roy, Jérôme Vicogne, Xuefen Le Bourhis, Roland P. Bourette

**Affiliations:** 1University of Lille, CNRS, Inserm, CHU Lille, Institut Pasteur de Lille, U1019-UMR 9017-CIIL-Center for Infection and Immunity of Lille, F-59000 Lille, France; 2University of Lille, CNRS, Inserm, CHU Lille, UMR9020-UMR1277—CANTHER—Cancer Heterogeneity, Plasticity and Resistance to Therapies, F-59000 Lille, France; 3Centre de Recherche en Cancérologie de Lyon, Department of Cancer Initiation and Tumor Cell Identity CNRS UMR5286, Inserm U1052, Lyon 1 University, F-69000 Lyon, France; 4University of Lyon, Universiteé Claude Bernard Lyon 1, CNRS UMR-5305, Laboratoire de Biologie Tissulaire et Ingeénierie Theérapeutique, F-69367 Lyon, France; 5Institut Pasteur de Lille—PLEHTA (Plateforme d’Expérimentation et de Haute Technologie Animale), F-59019 Lille, France; 6CNRS, Inserm, CHU Lille, Institut Pasteur de Lille, US41-UMS 2014-PLBS, University of Lille, F-59000 Lille, France

**Keywords:** M-CSF, CSF-1 receptor, prostate, C2H cell line, TRAMP, osteopontin, SPP1, transcriptome

## Abstract

Prostate cancer is a major public health concern and one of the most prevalent forms of cancer worldwide. The definition of altered signaling pathways implicated in this complex disease is thus essential. In this context, abnormal expression of the receptor of Macrophage Colony-Stimulating Factor-1 (M-CSF or CSF-1) has been described in prostate cancer cells. Yet, outcomes of this expression remain unknown. Using mouse and human prostate cancer cell lines, this study has investigated the functionality of the wild-type CSF-1 receptor in prostate tumor cells and identified molecular mechanisms underlying its ligand-induced activation. Here, we showed that upon CSF-1 binding, the receptor autophosphorylates and activates multiple signaling pathways in prostate tumor cells. Biological experiments demonstrated that the CSF-1R/CSF-1 axis conferred significant advantages in cell growth and cell invasion in vitro. Mouse xenograft experiments showed that CSF-1R expression promoted the aggressiveness of prostate tumor cells. In particular, we demonstrated that the ligand-activated CSF-1R increased the expression of *spp1* transcript encoding for osteopontin, a key player in cancer development and metastasis. Therefore, this study highlights that the CSF-1 receptor is fully functional in a prostate cancer cell and may be a potential therapeutic target for the treatment of prostate cancer.

## 1. Introduction

Prostate cancer is the most frequently diagnosed cancer in men worldwide. It represents a major public health concern in Western countries, and its incidence is increasing rapidly in Asia [1]. Despite major advances in its treatment, prostate cancer remains the leading cause of cancer death among men in 48 counties, including many developing countries in sub-Saharan Africa and Central and South America [2]. Since the androgen-signaling axis plays a pivotal role in this pathogenesis, androgen deprivation therapy (ADT) has been the basis of therapeutic strategies against locally advanced and metastatic prostate cancer. Although halted by prostatectomy and ADT, the disease often recurs as aggressive castration-resistant prostate cancer (CRPC) with capacities of invasion and metastasis. The emergence of this aggressive phenotype involves a variety of mechanisms, especially the interaction between cancer cells and the tumor microenvironment [3]. Since the pejorative prognosis of prostate cancer is mainly linked to its metastatic potential and therapies resistance, it is then important to understand the molecular events underlying these processes and to identify biomarkers of metastatic CRPC [3].

Receptor tyrosine kinases (RTKs) are key regulators of normal cellular processes since they lie at the head of a signal transduction cascade that modulates cell survival, proliferation, differentiation, adherence, and migration. Consequently, uncontrolled/biased RTK signaling plays a critical role in the majority of cancers, which made many RTKs attractive targets for cancer therapy [4,5], including for advanced prostate cancers [6].

The macrophage colony-stimulating factor (M-CSF or CSF-1) receptor (CSF-1R), encoded by the c-*fms* proto-oncogene [7,8], belongs to the RTK of class III that includes Kit, Flt3, and platelet-derived growth factor (PDGF) receptors [9]. This receptor binds both CSF-1 and interleukin 34 [10,11]. CSF-1R/CSF-1 receptor/ligand pair has essential physiological functions in monocyte/macrophage and osteoclast for proliferation and differentiation [12,13], which makes them of therapeutic interest [14,15]. The binding of CSF-1 induces CSF-1R dimerization and its autophosphorylation onto specific tyrosyl residues of the cytoplasmic domain. This creates binding sites for Src-homology 2 (SH2)-containing proteins such as Src, p85 subunit of PI3 Kinase, and Grb2 adapter, which in turn initiate multiple intracellular signaling pathways cooperating to regulate gene expression and, ultimately, cell survival, proliferation, and differentiation [16,17].

In addition to their role as an important regulator of hematopoiesis and bone resorption, CSF-1 and CSF-1R play a role in numerous cell types of non-hematopoietic origin. CSF-1R is expressed in normal placental trophoblast epithelium, and its activation by the locally high levels of CSF-1 produced by the endometrial epithelium is essential for normal embryonic implantation and placental development [18]. CSF-1R is also expressed in microglia and neural progenitor cells participating in central nervous system development [19]. CSF-1 and its receptor play a role in normal breast tissue during puberty, pregnancy, and lactation [13,20] and in breast carcinogenesis [21,22].

In normal mouse prostate, CSF-1R expression is restricted to early development. Though, CSF-1R expression has been reported in mouse and human prostate cancer tissues and cell lines [23,24]. This reexepression in adult cancer cells suggested a potential role of CSF-1R in prostate carcinogenesis. However, the functionality and biological effects of CSF-1R in prostate cells have not been investigated yet, unlike in other types of cancers [25,26]. Therefore, this study has endeavored to establish whether CSF-1R is functional in prostate cancer cells. Here we show that CSF-1R is functionally active in mouse and human prostate cancer cells. In the presence of CSF-1, CSF-1R activation induced tyrosine phosphorylation of intracellular signaling pathways and elicited proliferation and biological response. Most noticeably, CSF-1R expression greatly increased in vivo xenograft tumor growth. Moreover, we showed that the expression of numerous genes was increased by CSF-1 in prostate cancer cells, including the *spp1* gene that encodes for osteopontin, a major actor in prostate carcinogenesis.

## 2. Results

### 2.1. CSF-1 Receptor Is Functional in Murine Prostate Cancer Cells

To investigate the effects of CSF-1 receptor (CSF-1R or Fms) expression and activation in prostate cancer cells, the murine C2H prostate tumor cell line was used as a model. This clonal epithelial cell line was established from a prostate tumor of the transgenic adenocarcinoma mouse prostate (TRAMP) model [27]. The C2H cell line expresses cytokeratins, the androgen receptor, and is tumorigenic [28]. First, we transfected these cells with a firefly luciferase-expressing vector for further xenograft experiments. Thus, three stable luciferase-transfected C2H clones were obtained and called H9, N10, and N18 subclones. Transcripts of CSF-1 were detected by RT-PCR in C2H cells (Figure 1A). CSF-1 protein was detected in the culture medium conditioned by C2H parental cells and its three subclones, demonstrating that these cell populations produced and secreted CSF-1R ligand (Figure 1B). Although the C2H cell line derived from TRAMP tumors that express CSF-1R protein [23] and that Fms transcript was detected in these cells (Figure 1A), CSF-1R protein was not detected in these cell populations, neither by flow cytometric (Figure 1C) nor by western blot analyses (Figure 1D). Thus, murine c-fms cDNA was transduced in the H9, N10, and N18 subclones using a retrovirus vector [29]. Infected cells were sorted twice by flow cytometry to obtain cell populations expressing similar CSF-1R protein expression levels. This was assessed by flow cytometric (Figure 1C) and western blot (Figure 1D) analyses. Both CSF1-R isoforms (namely the gp140 Fms precursor and the gp165 Fms mature receptor) were detected, demonstrating the correct maturation of the CSF-1R as compared to FCP1-Fms myeloid progenitors and Raw264.7 macrophages (Figure 1D). Proteolytic cleavage of the CSF-1R [30] appeared to generate a fragment higher in prostate cells than in myelomonocytic FDCP1-Fms cells with the production of a 50–55 kDa intracellular fragment (Figure 1D). In order to assess its functionality, CSF-1R signaling was analyzed after stimulation by recombinant murine CSF-1. Myeloid FDCP1-Fms cells were used as a positive control [29]. Upon CSF-1 treatment, we clearly observed the autophosphorylation of the receptor and the activation of its two main canonical signaling pathways with phosphorylation of PKB/AKT and MAPK/ERKs (Figure 1E). Altogether, these results demonstrated that CSF-1R was functional in the prostate cellular environment.

### 2.2. CSF-1 Receptor Expression Promotes Prostate C2H Cell Growth and Invasion

We then evaluated the effect of CSF-1R expression on cell growth in the presence or absence of exogeneous CSF-1. As shown in Figure 2A, the addition of CSF-1 had no effect on the growth and morphology of H9, N10, and N18 cell parental clones that did not express CSF-1R. On the contrary, growth was significantly increased in CSF-1R-expressing cells, although no exogeneous CSF-1 was added to the culture medium, suggesting an autocrine activation of the receptor. Surprisingly, the addition of exogeneous CSF-1 significantly decreased cell growth (Figure 2A). This slowdown of growth was concomitant with a detachment of the cells from the substratum. Cells lost their flattened morphology, acquiring a rounded cell appearance (Figure 2B) which is similar to that of cells immediately after plating. Since all three clones displayed very similar responses in regard to cell growth, one clone was chosen to investigate cell invasion in response to CSF-1. The invasive potential of N18 and N18-Fms cells was determined by measuring invasion through a barrier of the reconstituted basement membrane, Matrigel, over a 72 h-period in the presence or absence of exogenous CSF-1 (Figure 2C). N18 cells were not able to invade through Matrigel with or without CSF-1. Similarly, N18-Fms cells did not show invasive activity in the absence of CSF-1 (Figure 2C). However, after the addition of exogeneous CSF-1, N18-Fms invasive activity was significantly increased (Figure 2C). Altogether, these data showed that CSF-1R expression in prostate cancer cells altered their potential in terms of growth, adherence, and invasion. Since all these biological properties play key roles in tumor development, this strongly suggests that CSF-1R has the ability to stimulate prostate cancer progression.

### 2.3. CSF-1 Receptor Expression Increased Tumor Cell Growth In Vivo

We next tested both parental and CSF-1R-expressing C2H subclones for their growth in vivo after subcutaneous injection in male immunodeficient SCID mice. The results obtained with the cells inoculated in both flanks, either as parental cells or those expressing the CSF-1R, are shown in Figure 3. Two subclones were selected for tumor growth monitoring, each of them with a different technique. First, H9 and H9-Fms cell growth was monitored by non-invasive bioluminescence imaging (BLI). After an initial latency, H9-Fms cells displayed rapidly growing tumors, illustrated by a considerable increase in radiance (Figure 3A). In contrast, parental H9 cells demonstrated a long lag phase before producing a slight increase in radiance (Figure 3A). Figure 3B shows the progression of tumor burden by BLI analysis of two representative mice at two different time points (17 and 31 days after cell injection). The positive effect of CSF-1R expression on tumor growth was confirmed in a second experiment using N18 and N18-Fms cells (Figure 3C). Measurements of xenograft tumors collected with manual calipers showed that N18 control cells produced small tumors after a long lag phase (Figure 3C,D). In contrast, the rapid development of tumors was observed after N18-Fms injection (Figure 3C,D).

### 2.4. Transcriptome Analysis Discovered Key Functions Regulated by CSF-1R in Murine Prostate Cancer Cells

We next compared the transcriptome profiling of prostate cancer cells stimulated or not by CSF-1 for 24 h or 48 h. For this purpose, the N18-Fms cells were chosen. The analysis of the 60K Agilent microarrays revealed 1258 and 1117 mRNA were significantly differently expressed between unstimulated cells and 24 h or 48 h CSF-1-stimulated cells, respectively (Figure 4A,B). A total of 546 genes were both upregulated after 24 h and 48 h of CSF-1 stimulation as compared to unstimulated cells (Figure 4B, upper panel), and 194 genes were both downregulated after 24 h and 48 h upon CSF-1 stimulation as compared to unstimulated cells (Figure 4B, lower panel).

Genes upregulated in both 48 h and 24 h upon CSF-1 treatment when compared to untreated control (as shown in Figure 4B upper panel) were subjected to a gene ontology enrichment analysis using the GO Biological Process (Figure 4C, left panel) and Reactome terms (Figure 4C, right panel). Several immune-related biological processes, including regulation of viral genome replication, Jak-STAT signaling pathway, inflammatory response, cytokine, and interferon signaling, were significantly enriched in CSF-1 stimulated cells. This suggested that some CSF-1R functions in macrophages were conserved in prostate cancer cells. Interestingly, gene ontology enrichment analysis of genes downregulated revealed the enrichment in the cell adhesion process and negative regulation of epithelial cell proliferation (Figure 4D, left panel). This result was then in agreement with the biological effect of CSF-1 that we observed in prostate cancer cells stimulated by exogenous CSF-1 (Figure 2 and Figure 3).

An overview of all enriched biological categories and gene pathways revealed in the analysis is given in Appendix A. Notably, several genes that were previously reported to be associated with cancer development were changed in their expression after CSF-1 stimulation. Among them, spp1 encoding for osteopontin was significantly upregulated both after 24 h and 48 h of CSF-1 stimulation (Figure 4A). This prompted us to investigate the expression of spp1/osteopontin in CSF-1-stimulated prostate cancer cells.

### 2.5. Spp1 Gene Encoding Osteopontin Is a Target of CSF-1R in Murine Prostate Cancer Cells

Osteopontin (OPN) is a secreted protein produced by numerous cell types, including macrophages [31]. Since CSF-1 is the primary regulator of macrophages, this prompted us to test if ligand-activated CSF-1R is regulating spp1 gene expression in macrophages. First, using the CSF-1-dependent macrophage cell line Bac1.2F5 (Figure 5A), we measured a net increase in the expression of osteopontin transcripts in response to CSF-1 (Figure 5B). Next, we investigated the expression of the spp1 gene in the three different prostate subclones stimulated by CSF-1. As shown in Figure 5C, CSF-1 induced a significant increase in osteopontin transcript in the three CSF-1R-expressing clones but not in the control cell populations that did not express the CSF-1R. To extend these results to two other cell lineages, we tested the C3-223 mammary cancer cell line and the NIH 3T3 fibroblast cell line where CSF-1R was overexpressed similarly to prostate C2H subclones (Figure 5D). CSF-1 stimulation increased spp1 gene expression only in the C3-223-Fms mammary cancer cells (Figure 5E) but not in the NIH 3T3. Together, these results showed that spp1/osteopontin is a target gene of the CSF-1/CSF-1R axis in both prostate and breast cancer cells but not in the NIH3T3 fibroblast cell line.

### 2.6. CSF-1R Is Functional in Human Prostate Cancer Cells and Increases Osteopontin Gene Expression

To determine if CSF-1R is also functional in human prostate cancer cells, we selected two malignant prostate cancer cell lines. M12 is a subline derived from the P69 prostate cell line and displays a mesenchymal-like, highly tumorigenic, and metastatic phenotype [32,33]. Interestingly, we observed that M12 cells express low levels of CSF-1R on the cell surface (Figure 6A, left panels). PC3 is an androgen-independent cell line derived from bone metastasis [34]. No CSF-1R expression could be detected on the cell surface (Figure 6A, left panels). Human CSF-1R was then overexpressed in both cell lines leading to a cell surface expression similar to those obtained with C2H subclones (Figure 6A, right panels). After CSF-1 stimulation of M12-Fms cells, we observed tyrosine phosphorylation of the CSF-1R, AKT, and ERKs, demonstrating the functionality of the CSF-1R in these cells (Figure 6B). In PC3-Fms cells, CSF-1 stimulation properly induced autophosphorylation of its receptor (Figure 6B). AKT was highly phosphorylated in the absence of CSF-1, and no detectable increase in AKT phosphorylation was observed in response to CSF-1 (Figure 6B). Although CSF-1R is functional in these cells, no ERK phosphorylation was observed in response to CSF-1 (Figure 6B). These results were similar to those demonstrating that in PC3 cells, the absence of PTEN induced constitutive AKT activity, leading to the inactivation of the MAPK/ERK pathway [35]. In culture, the addition of CSF-1 modified PC3-Fms cell morphology to a more rounded shape (Figure 6C). Finally, CSF-1 induced a significant increase in spp1/osteopontin gene expression in both Fms-expressing PC3 and M12 cells but not in the PC3 control cells that do not express the CSF-1R (Figure 6D). A low but significant induction was observed in M12-Ctl cells, probably due to low cell surface Fms expression in this cell line (Figure 6A). All together and in agreement with those obtained in murine cells, these results confirmed the functionality of CSF-1R in prostate cancer cells and osteopontin as a downstream target of CSF-1 signaling in these cells.

## 3. Discussion

Abnormalities in RTK expression and activation are associated with a wide range of diseases. In cancer, RTK plays a key role in the development, progression, and resistance to therapy [5]. Of particular interest, multiple RTKs and their respective ligand are implicated in complex signaling pathway crosstalks supporting both tumor cell metastasis and resistance to therapy [36]. A large number of studies aimed to dissect the RTK signaling pathways and the consequences of their inhibition on tumor development. In the case of the CSF-1/Fms receptor, two aspects are relevant to tumor biology. First, CSF-1R regulates tumor-associated macrophages (TAM) to promote tumor progression. TAM represents a major part of immune cell populations that infiltrate into tumors and contribute to tumor progression at multiple levels, including immunosuppression. CSF-1R is thus a prime target for treating solid tumors such as prostate or breast cancer alone or in combination with chemotherapy [37] or radiation therapy [38]. Though targeting CSF-1R expression on TAM appears as a promising strategy [39], the beneficial effects still need to be further investigated [40,41,42]. Second, numerous studies have demonstrated that CSF-1R also drives tumor cell progression in several cancer types, including renal cell carcinoma [43], ovarian cancer [44], colorectal cancer [45], bladder cancer [46], mesothelioma [47], and melanoma [48]. Breast cancer is the most consistently documented type of cancer concerning the expression and function of CSF-1R. Its expression has been demonstrated in various breast cancer cell lines and tissues specimen. In breast carcinoma cells, CSF-1R has been demonstrated as being functional and implicated in various biological effects, such as proliferation, invasion, and epithelial-mesenchymal transition [21,25,49,50,51,52,53,54,55]. Moreover, its expression is correlated with poor clinical outcomes [22,56]. Thus, the role of the CSF-1/CSF-1R axis in supporting the growth and motility of several solid tumors has been well documented [57].

Few studies have investigated the expression of CSF-1R and its ligand in prostate cancer. Ide et al. [23] described CSF-1R expression in the adenoma of the prostate of the TRAMP mouse model of prostate cancer [27] and in human prostate cancer cell lines. Using a panel of the human prostate cancer tissue specimen, they showed by immunohistochemistry that CSF-1R is expressed by almost all specimens with the most intense signal in prostatic intraepithelial neoplasia (PIN) and carcinomas of histological Gleason grade three or four. Richardsen et al. [58] showed that both CSF-1R and CSF-1 were expressed in prostate tumors with higher expression in patients with metastatic prostate cancer as compared to patients with non-metastatic prostate cancer. Yet no study has been performed that specifically aimed at determining the biological effects of CSF-1R in prostate cancer cells. This prompted us to explore whether CSF-1R expression is functional in prostate cancer cells.

Here we have shown that in response to CSF-1, CSF-1R autophosphorylated and activated two major signaling pathways, ERK1/2 and AKT. Thus, signaling pathways downstream of CSF-1R in prostate cancer cells were similar to those in myelomonocytic cells [16]. Interestingly, tyrosine phosphorylation patterns after CSF-1 stimulation were similar but slightly different between myeloid FDCP1-Fms cells and prostate C2H-Fms cells (Appendix A), suggesting that tissue specificity for the CSF-1R substrates may exist. These notable differences warrant further research to decipher the specific substrates of CSF-1R in carcinoma. A previous analysis of the downstream substrates of CSF-1R in breast cancer cells has unveiled novel CSF-1R targets, which could then play a specific role in epithelial tumorigenesis [50]. Expression of CSF-1R alone was sufficient to increase cell proliferation in monolayer culture, suggesting an autocrine stimulation of these cells, which produce CSF-1. Upon addition of CSF-1 to the culture medium, the cellular response was modified with a decrease in cell proliferation and a change from flattened to rounded cell morphology with a trend to detachment. The addition of CSF-1 also increased cell invasiveness in the presence of a matrix gel. This suggested that CSF-1 concentration could influence the nature of cell response. Interestingly, we have previously demonstrated that strong CSF-1R stimulation was required in myelomonocytic cells to obtain a differentiation response, with a central role in MAPK pathway activation [59]. Similarly, in lung cancer cells, CSF-1 level and duration of stimulation controlled the cell invasiveness [60]. Thus, a local environment with high CSF-1 concentration may favor tumor cells to invade and metastasize. Tumor stroma contains several cell types capable of producing CSF-1, such as endothelial cells and fibroblasts. However, the first candidates to consider are TAM, which produces CSF-1 and is implicated in most of the steps of tumoral development, especially invasion [61,62]. Macrophages may also favor carcinoma cell invasion through a CSF-1/epidermal growth factor paracrine loop, as described in breast cancer [63]. Here we have shown that CSF-1R expression had a tremendous effect on tumor growth as compared to cells that did not express CSF-1R. Altogether, these results strongly suggest that CSF-1R is able to play an active role during prostate cancer development, which is in accordance with the previous studies demonstrating that CSF-1R expression is correlated with poor prognosis of prostate cancer [23,58]. A recent study by Kwon et al. using transgenic mice expressing CSF-1 in the prostate gland demonstrates that forced CSF-1 expression promotes immune cell infiltration and low-grade PIN but fails to transform prostate cells [64]. Interestingly, CSF-1R was not detected in prostate cells in these PIN lesions, suggesting that CSF-1R expression might be an important step in tumor progression toward carcinoma. The mechanisms of abnormal CSF-1R in all the various types of tumors, prostate, breast, and ovary, still remain unknown and warrant further investigations.

While conducting cDNA microarray assays for differentially expressed genes downstream of CSF-1R signaling, we identified spp1/osteopontin as a gene of interest. Osteopontin is a multifunctional phospho-glycoprotein secreted in the microenvironment and a key regulator of tumor progression and immunomodulation [65,66]. We showed that osteopontin transcript expression was increased by CSF-1 both in murine and human prostate cancer cells. Interestingly, we showed that the *spp1* gene was a CSF-1R-responsive gene in a macrophage cell line, a mammary cell line, but not in a fibroblast cell line. This suggests that spp1/osteopontin is one example of CSF-1-responsive genes shared between myelomonocytic lineage and carcinoma cells expressing CSF-1R. Considering the complex relationship between CSF1/CSF-1R-expressing carcinoma cells and macrophages, synthesis of OPN is of particular interest since OPN is a chemoattractant for macrophage and regulates its adhesion, migration, differentiation, and production of cytokine [65,67,68]. Thus, Zhu et al. have shown that OPN stimulates the synthesis of CSF-1 by TAM of hepatocellular carcinoma and identifies the OPN/CSF-1/CSF-1R axis as a critical mediator in the immunosuppressive nature of the cancer microenvironment [69]. In prostate cancer, numerous pieces of evidence have shown that OPN is closely associated with the proliferation and metastasis of cancer cells [70]. In prostate cancer mouse models, OPN expression was observed at all different levels of tumor progression and increased from PIN to adenocarcinoma, with the highest level during metastatic progression [71]. OPN has been described as a biomarker and a potential therapeutic target for metastatic castration-resistant prostate cancer [72]. Messex et al. have recently shown that macrophage-produced OPN that stimulates the growth of prostate cancer cells [73]. Thus, OPN is one example of how prostate cancer cells, via the CSF-1/CSF-1R axis, are able to remodel the extracellular matrix and the immune landscape.

In summary, this study has demonstrated the functionality of CSF-1R in prostate cancer cells and has unveiled osteopontin as a CSF-1R-target gene. Given that CSF-1R also plays a critical role in the tumor immune landscape, more research is warranted in this area for a better understanding of how CSF-1R regulates prostate cancer development and how this knowledge could be used in terms of therapeutic strategies.

## 4. Materials and Methods

### 4.1. Cell Lines and Cell Culture

TRAMP-C2 cell lines were established from the spontaneous TRansgenic Adenocarcinoma of Mouse Prostate (TRAMP) model [27]. TRAMP C2H clonal cell line, a gift of Dr. Norman Greenberg, was established by three rounds of dilutional cloning from the previously characterized tumorigenic TRAMP-C2 cell line [28]. TRAMP C2H cells were transfected with pGL4.50 [luc2/CMV/hygro] or pGL4.50 [luc2/CMV/neo] (Promega, Charbonnières-les-bains, France) using PEI/ExGen 500 (Euromedex, Souffelweyersheim, France), according to the manufacturer’s instructions. Transfected cells were selected using either hygromycin (Sigma, Saint-Quentin-Fallavier, France; 250 μg/mL) or neomycin (Sigma, G418, 1 mg/mL), and individual clones were isolated by limiting dilution and tested for luciferase activity using dual-luciferase reporter assay system (Promega). Two C2H neomycin-resistant clones (N10 and N18) and one hygromycin-resistant clone (H9) were selected for further analysis. Cells were maintained in Gibco high-glucose Dulbecco’s modified Eagle’s medium (DMEM, Invitrogen, Illkirch-Graffenstaden, France) containing 10% fetal bovine serum (FBS, Invitrogen) supplemented with antibiotics (Zell Shield, Biovalley, Illkirch-Graffenstaden, France) and dihydrotestosterone (100 μM, Androstan, Sigma, #A8380) at 37 °C in a humidified atmosphere containing 5% CO_2_. The murine hematopoietic interleukin-3 (IL3)-dependent cell line FDCP1-Fms was maintained in DMEM-10% FBS supplemented with IL3 and antibiotics [29]. NIH 3T3 mouse fibroblasts and C3-223 mouse mammary cancer cells were maintained in DMEM-10% FBS supplemented with antibiotics. The C3-223 cell line has been established from the transgenic FVB C3(1)-Tag mouse model of breast cancer [74]. Briefly, a mammary tumor was dissociated, and single cells were placed in culture in a 10 cm-plastic dish (Corning, Dutscher, Bernolsheim, France). Adherent cells were kept in culture for up to 40 passages (5 months) without any sign of senescence. C3-223 cells are of epithelial origin, as evidenced by T antigen expression (Appendix A) that occurred in the mammary epithelium of C3(1)-Tag female mice [75]. The human M12 cell line [32] was a gift from Dr. BS Kundsen (Fred Hutchinson Cancer Research Center, Seattle, USA). The human PC3 cell line was obtained from American Type Culture Collection. Both human cell lines were maintained in Gibco Roswell Park Memorial Institute -1640 medium (RPMI, Invitrogen) supplemented with 10% FBS and antibiotics.

### 4.2. CSF-1R Transduction

Murine wild-type Fms receptor was introduced in different murine cells by retroviral infection using 0.45 μm-filtered supernatants of packaging Psi-2 Fms cells in the presence of polybrene (4 μg/mL) as previously described [29]. Infected cells were selected twice for cell surface Fms expression by FACS Aria II (BD Biosciences, Le Pont de Claix, France) after staining with anti-mouse CD115 antibody, APC (Thermo Fisher Scientific, Illkirch-Graffenstaden, France, eBioscience #17-1152-80). Human PC3 and M12 cells were transfected with a human CSF-1R expression vector (GenScript, Piscataway, NJ, USA, #OHU24034) or empty pcDNA3.1 vector as a control, using JetPRIME transfection reagent, according to the manufacturer’s guidelines (Polyplus Transfection, Illkirch-Graffenstaden, France). Transfected cells were selected using G418 (0.5 mg/mL), then cells transfected with CSF-1R expression vector were further selected for cell surface Fms expression by FACS Aria II (BD Biosciences) after staining with anti-human CD115 antibody, PE (eBioscience, #12-1159-42).

### 4.3. Growth Studies

Cell proliferation was evaluated by plating 2 × 10^4^ cells per plate in a 6-well plate in 2 mL of culture medium with or without 50 ng/mL of recombinant murine CSF-1 (rmCSF-1) (#315-02, Peprotech, Neuilly-sur-Seine, France). After 72 h, cells were isolated after trypsinization using trypsin–versene (EDTA) solution, and cell counting was performed under the microscope using a Malassez counting chamber.

### 4.4. Invasion Assay

Invasion assay was performed using inserts BD Falcon FluoroBlok (#351152, BD Biosciences) with 8 μm pore size PET membrane on 24-well insert companion plate (#353504, BD Biosciences). Inserts were coated with 50 μL of Matrigel^®^ matrix diluted to half in serum-free DMEM, then incubated for 1 h at 37 °C. For the experiment, each condition was performed in triplicate; 500 μL of DMEM-1% FBS containing 10^4^ cells were deposited on each insert. Inserts are introduced into insert companion plate containing 750 μL of DMEM-1% FBS with or without rmCSF-1 (50 ng/mL). After five days at 37 °C in 5% CO_2_ atmosphere, the invasive potential of cells was measured by fluorescence using FLUOstar OPTIMA plate reader (BMG Labtech, Champigny-sur-Marne, France) after cell staining with calcein AM (#65-0853, eBioscience) for 1 h at 37 °C.

### 4.5. ELISA

For CSF-1 quantification, 5 × 10^4^ cells per well were plated in 1 mL of culture medium in 6-well plates for 48 h. Conditioned medium was collected, filtered 0.45 μm, and subjected to ELISA using commercially available mouse M-CSF ELISA Kit (#900-K245, Peprotech) according to the manufacturer’s instructions. The CSF-1 secretion was measured with a standard curve.

### 4.6. RT-PCR

Total RNA was isolated using the RNeasy mini kit (Qiagen, Courtaboeuf, France). The QuantiTect reverse transcription kit (Qiagen) was used to convert RNA to cDNA as per manufacturer’s instructions. Standard PCR was performed in 0.2 mL thin-walled tubes (Thermo Fisher Scientific) using KAPA2G hot start kit (Kapa Biosystem, KK562). Thermal cycling was carried out in an MJ Research PTC-100 instrument (Watertown, MA) in a 25 μL reaction volume. PCR products were size-fractionated on a 1.5% ethidium bromide-containing agarose gel. For quantitative PCR (qPCR), diluted cDNAs were transferred to 96-well PCR optical plates (Axygen). KAPA SYBR FAST qPCR kit (Kapa Biosystems, Saint-Quentin-Fallavier, France) was used. qPCR was performed using the Agilent Mx3000P detection system (Agilent Technologies, Les Ullis, France). Relative mRNA levels were determined following normalization to mouse RPLP38 or human RPLP0 housekeeping genes and analysis of the comparative threshold cycle (2^−ΔΔCt^) method. Experiments were performed in duplicate. Primer sequences are presented in Table A1 (Appendix B)

### 4.7. Microarray Analysis

N18-Fms cells were stimulated by rmCSF-1 (100 ng/mL) for 12 h or 24 h or left unstimulated. Total RNA from two independent experiments was extracted with the RNeasy Plus Kit (Qiagen) and was quantified with the NanoDrop ND-1000 (Thermo Fisher Scientific). Quality of the extracted RNA was evaluated with the Agilent 2100 Bioanalyzer system (Agilent Technologies). cRNA labeling, hybridization, and detection were carried out according to supplier’s instructions (Agilent Technologies). For each microarray, Cyanine 3-labeled cRNA was synthesized with the low input Quick Amp WT labeling kit from 100 ng of total RNA. RNA Spike-In was added to all tubes and used as positive controls of labeling and amplification steps. Labeled cDNA samples (600 ng) were used for 17- hour hybridization at 65 °C to the SurePrint G3 mouse GE 8 × 60K Microarrays (Agilent Technologies). All procedures were carried out according to the manufacturer’s instructions. The hybridized microarrays were scanned with the Agilent DNA Microarray Scanner (G2505C). Signal intensity was quantified from the scanned image by using Feature Extraction software version 11.0.1 (Agilent Technologies). Statistical comparisons and filtering were achieved with the Genespring^®^ software version GX12.0 (Agilent Technologies). All these experiments were performed by Imaxio Company (Clermont- Ferrand, France). Further investigations were carried out using the GO biological process and Reactome pathways databases.

### 4.8. Immunoblotting and Antibodies

Prostate cell lines were starved overnight at 37 °C in DMEM-0.5% FBS medium. FDCP1-Fms cells were starved in DMEM-1% FBS for 3 h at 37 °C. Cells were washed in phosphate-buffered-saline (PBS, Thermo Fisher Scientific) and stimulated by 2500 U/mL of recombinant murine CSF-1 [29] or 500 ng/mL of recombinant human CSF-1 (Peprotech, #AF-300-25) for different times at 37 °C. They were then lysed with 100 μL RIPA buffer (20 mM HEPES, 1% NP40, 0.1% SDS, 5% glycerol, 142 mM KCl, 5 mM MgCl2, 1 mM EDTA, pH 7.45) supplemented with phosphatase inhibitors (1/200 Phosphatase Inhibitor Cocktail 2, Sigma # P5726) and protease inhibitors (1/400 Protease Inhibitor Cocktail, Sigma # P8340). Lysates were centrifuged and the supernatant collected. The total protein concentration was determined with the Pierce BCA Protein Assay Reagent Kit (Thermo Fisher Scientific #23225), and equal amounts of proteins (20–30 μg) were resolved on NuPAGE 4–12% Bis-Tris gels (Thermo Fisher Scientific, #NP0335BOX). Separated proteins were transferred onto a polyvinyl difluoride membrane in Towbin buffer (10% methanol, 10% Tris-glycine 1X, 0.0025% SDS). The membrane was then equilibrated in blocking buffer (casein 2 g/L, PBS 1X, 0.05% Tween). Proteins were analyzed by western blotting with anti-phosphorylated mouse/human M-CSF receptor (PhosphoTyr723, Ozyme, Saint-Cyr-L’école, France, Cell Signaling #3155), anti-phosphorylated-Akt (D9E) (Ser473, Cell Signaling #4060), anti-Akt (Cell Signaling #4691), anti-phosphorylated-p44/42 MAPK (Erk1/2) (Thr202/Tyr204, Cell Signaling^®^ #9106), anti-Erk2 (C14) (Santa Cruz, Heidelberg, Gerrmany, sc-154), and anti-actin (C4) (Santa Cruz, sc47778). After incubation with the appropriate species-specific horseradish-peroxidase-conjugated secondary antibodies (anti-rabbit (#711-035-152), ant-mouse (#115-035-146), or anti-goat (#705-035-003) (Jackson ImmunoResearch Lab^®^, Saint-Cyr-L’école, France), the antigen-antibody complexes were detected using Super Signal West Dura Extended Duration Substrate (Thermo Fisher Scientific, #34076). Luminescence was captured by digital imaging with a cooled-charge-coupled device camera (LAS 4000, Fuji, Tokyo, Japan). For all immunoblot images presented throughout this manuscript, the membranes were cut into horizon pieces according to the estimated molecular weight of proteins of interest and probed with the indicated antibodies. All cropped blots were run under the same experimental condition.

### 4.9. Tumor Xenograft Assay

Eight-week-old male severe combined immunodeficient (SCID) mice were purchased from Pasteur Institute of Lille (Lille, France) and were maintained in accordance with the Institutional Animal Care and Use Committee procedures and guidelines. Further, 1.5 × 10^5^ C2H-derived clones were resuspended in 150 μL of PBS before subcutaneous injection into flanks, with Fms-expressing cells on the left flank and control cells on the right flank of the same mice (n = 8). Tumor volume was calculated as follows: tumor volume = width × length × (width + length)/2. In order to determine the tumor volume using a bioluminescence imager (IVIS Lumina XR Caliper LS, PerkingElmer, Villebon-sur-Yvette, France), each mouse was monitored at the indicated time after intraperitoneal injection of luciferase substrate luciferin (D-luciferin Firefly potassium salt, Caliper Lifesciences, Villepinte, France, #122796) at 150 mg/kg body weight.

### 4.10. Statistical Analysis

Data are presented as mean values ± standard deviation (s.d.) of at least 3 independent experiments. The statistical analysis was performed by using a 2-sided Student *t*-test; *p* values < 0.05 were considered to be significant.

## Figures and Tables

**Figure 1 ijms-23-16028-f001:**
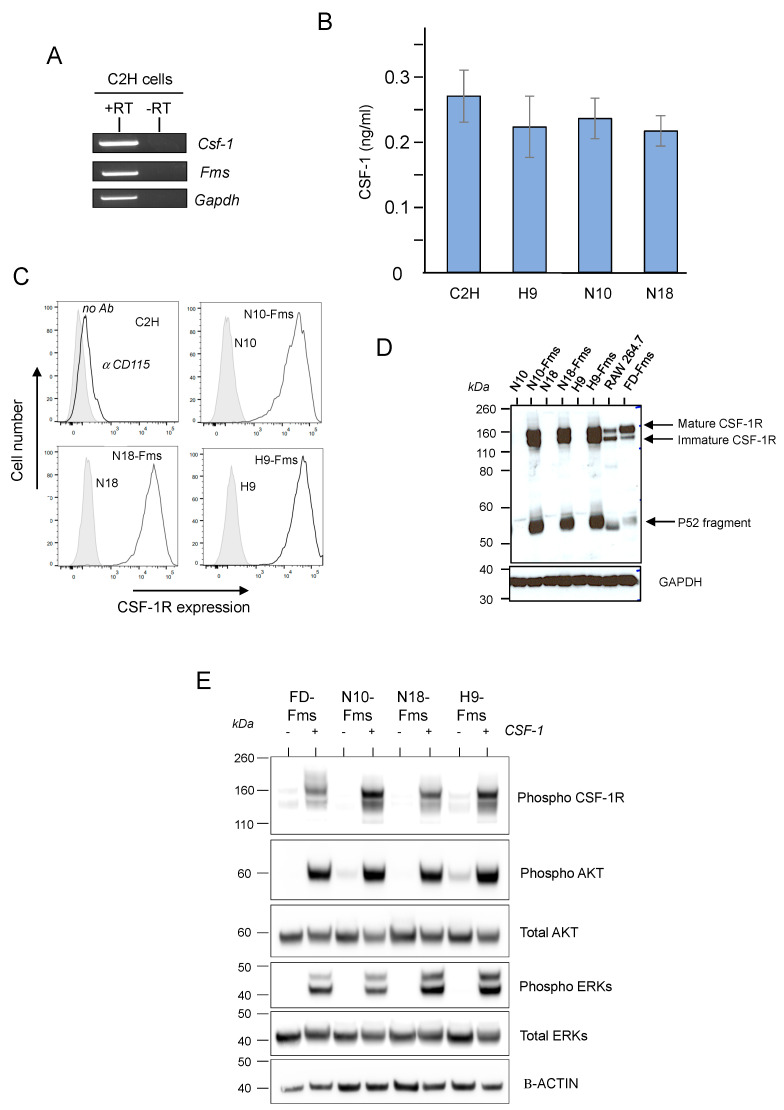
The CSF-1 receptor is functional in C2H murine prostate cells and activates major signaling pathways. (**A**) RT-PCR analysis of CSF-1, CSF-1 receptor (Fms) and GAPDH transcripts in C2H cells. N = 3. Reverse transcriptase (RT) (**B**) ELISA assay to quantify CSF-1 secretion by C2H parental cells and three subclones, H9, N10, and N18. N = 4 (**C**) murine c-*fms* cDNA encoding CSF-1R was trans-duced in the three subclones by retroviral infection; CSF-1R-expressing cells were isolated by FACS and three cell populations were obtained, called H9-Fms, N10-Fms, and N18-Fms. Cell surface expression was analyzed by flow cytometry. (**D**) Immunoblot shows that CSF-1R protein was expressed and matured as compared to Raw 264.7 and FDCP1-Fms (FD-Fms) myelomonocytic cell lines. (**E**) Western blot analysis shows that CSF-1 stimulation of Fms-expressing prostate cells activates receptor kinase activity and phosphorylation of AKT and p42/p44 ERKs substrates. As a control, activation of myeloid FD-Fms cells by CSF-1 shows similar AKT and ERKs phosphorylation. For all immunoblot images presented throughout this manuscript, the membrane was cut into pieces according to the estimated molecular weight of proteins of interest and probed with the indicated antibodies. All cropped blots were run under the same experimental conditions. Data presented here are representative of at least 3 independent experiments.

**Figure 2 ijms-23-16028-f002:**
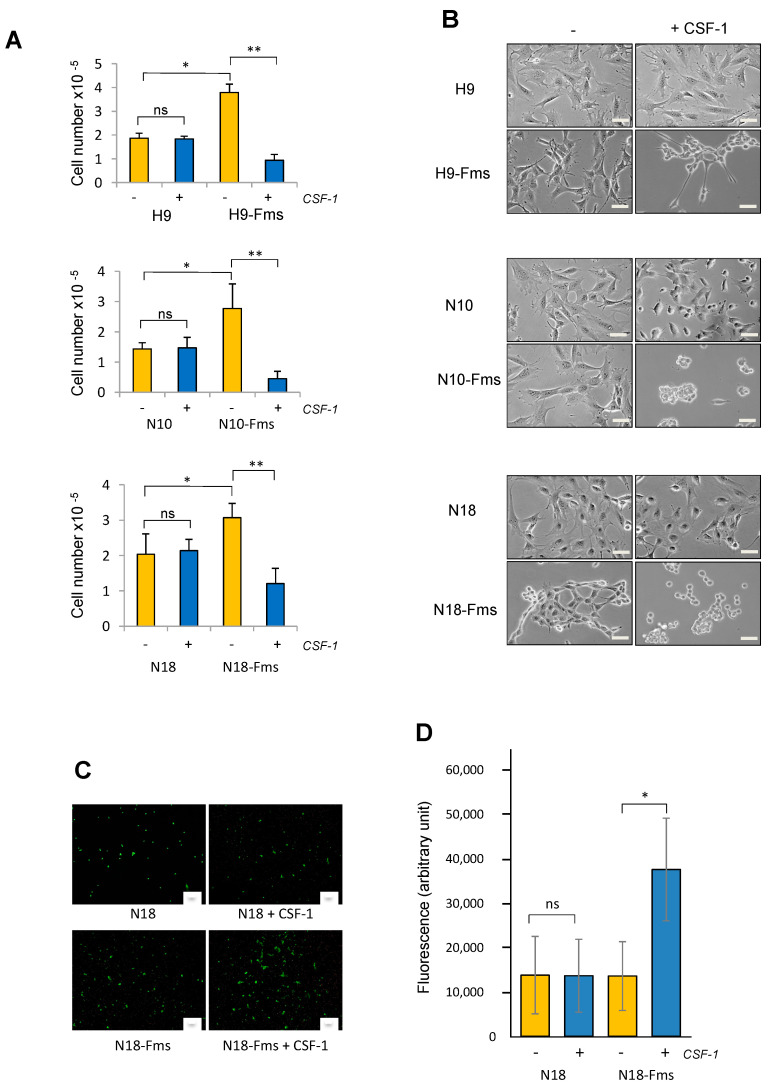
Biological effects of CSF-1R signaling on prostate cancer cells. (**A**,**B**) Effects on cell growth in 2D culture: 2 × 10^4^ prostate cells were seeded in the presence (+) or absence (-) of CSF-1 for 3 days and enumerated. (**B**) Representative photographs (n = 3) of each cell populations after 3 days of culture in the presence (+) or absence (-) of CSF-1, scale bars, 10 μm. (**C**,**D**) Effects on cell invasion. Invasive capacities of parental N18 prostate cells and CSF-1R-expressing N18 prostate cells (N18-Fms) were determined by transwell assay in the presence (+) or absence (-) of CSF-1. (**C**) Representative photographs (n = 3) of each cell populations after 5 days of culture in the presence (+) or absence (-) of CSF-1; cells were stained with calcein AM. Scale bars, 100 μm. (**D**) Fluorescence values are expressed as mean ± s.d of 3 independent experiments. ns = not significant, * *p* < 0.05, ** *p* < 0.01.

**Figure 3 ijms-23-16028-f003:**
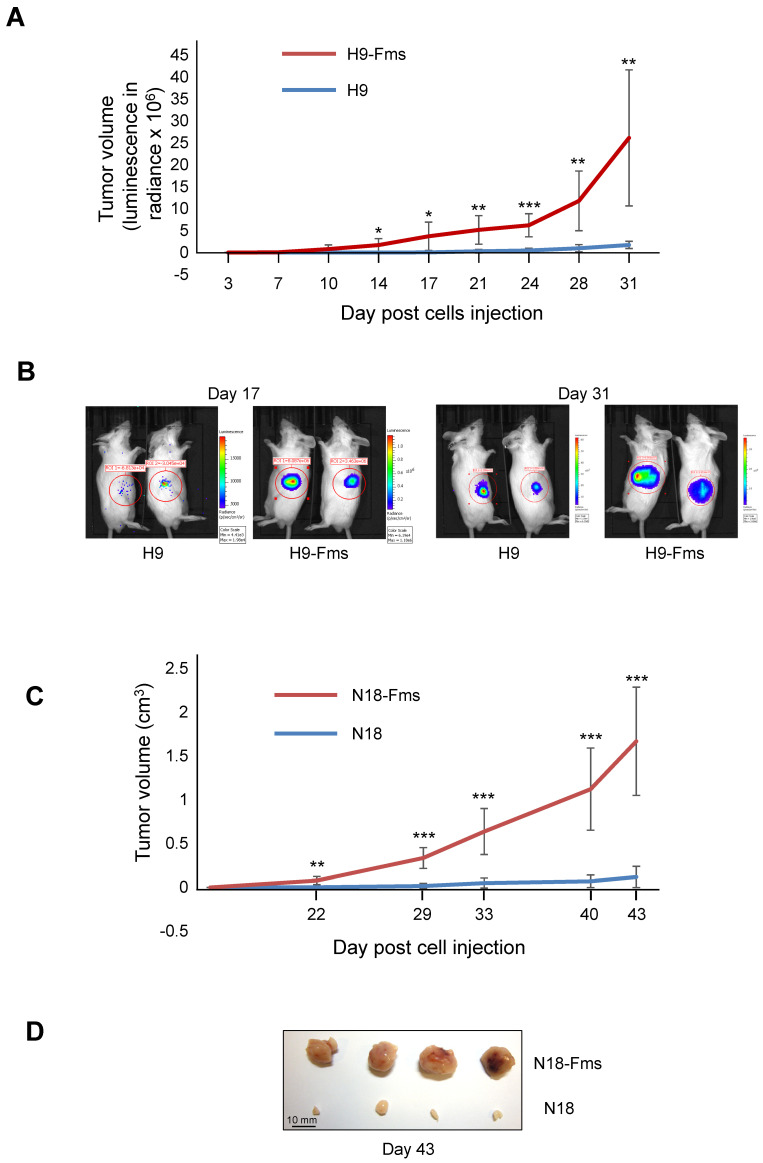
Tumor volume evolution of xenografted prostate C2H subclones. (**A**) Each male SCID mice were injected subcutaneous with H9-Fms (left flank) and control H9 (right flank) cells, both cell populations stably expressing luciferase gene. Every 3 or 4 days, luminescence was measured in radiance (photon/second/cm^2^/steradians). (**B**) Photographic demonstration of comparative tumor size of 2 mice at day 17 (left panels) and day 31 (right panels). (**C**) Each male SCID mice were injected subcutaneous with N18-Fms (left flank) and control N18 (right flank) cells. Tumor volume was measured using calipers. (**D**) Forty-three days after cell injection, tumors were removed from 4 mice and photographed. For all these experiments, each group contained eight mice. The values are expressed as mean ± s.d. * *p* < 0.05, ** *p* < 0.01, *** *p* < 0.001.

**Figure 4 ijms-23-16028-f004:**
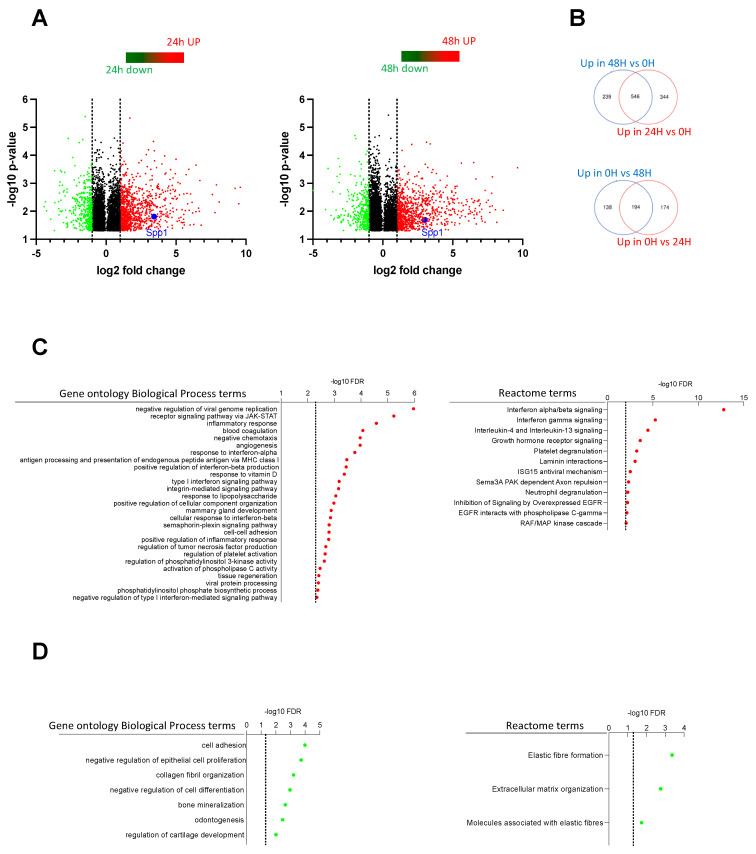
Transcriptome analysis of N18-Fms cells stimulated by CSF-1. (**A**) Volcano plots showing all genes differentially expressed with *p* < 0.05 when comparing 24 h (left plot) and 48 h (right plot) of CSF-1 treatment with untreated control (0 h) (N = 2 for each group). Genes upregulated and downregulated with a fold change of 2 (shown by the grey line) or more after treatments are shown by red and green dots, respectively. The *Spp1* gene is shown by a blue dot. (**B**) Venn diagram showing the intersection between genes upregulated (upper diagram) and downregulated (lower diagram) by the 24 h and 48 h CSF-1 treatments when compared to the untreated control. (**C**) Genes upregulated in both 48 h and 24 h of CSF-1 treatment when compared to untreated control (as shown in (**B**)) were subjected to a gene ontology enrichment analysis using the GO Biological Process (left panel) and Reactome terms (right panel). The −log10 of the FDR is shown for each enriched term with a FDR < 0.005 for Biological Process terms and a FDR < 0.01 for Reactome terms (shown by the dotted line). These thresholds were chosen for clarity purpose and the complete list of enriched terms is shown in Appendix A. (**D**) Same as in (**C**) using the genes downregulated in both 48 h and 24 h of CSF-1 treatment when compared to untreated control. All enriched terms with a FDR < 0.05 are shown.

**Figure 5 ijms-23-16028-f005:**
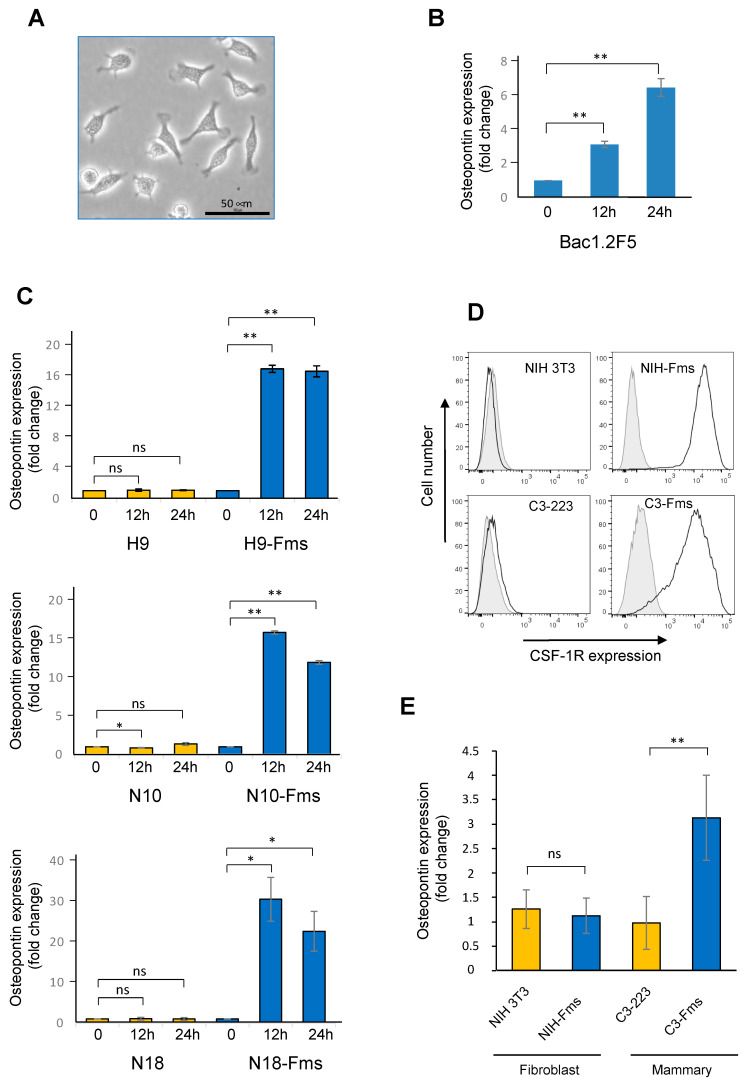
CSF-1 increases spp1/osteopontin mRNA expression in murine macrophages, prostate cancer cells and breast cancer cells. (**A**) Representative image (n = 3) of Bac1.2F5 murine macrophages cultured in the presence of CSF-1. (**B**) Relative expression of spp1/osteopontin mRNA was assessed by qRT-PCR performed in triplicate (normalized against RPLP38). Starved Bac1.2F5 cells were cultured or not in the presence of CSF-1 for 12 or 24 h. (**C**) Relative expression of spp1/osteopontin mRNA was assessed by qRT-PCR performed in triplicate (normalized against RPLP38). H9 and H9-Fms prostate cells (upper panel), N10 and N10-Fms prostate cells (middle panel), and N18 and N18-Fms prostate cells (lower panel) were cultured or not in the presence of CSF-1 for 12 or 24 h. (**D**) murine c-*fms* cDNA encoding CSF-1R was transduced in murine fibroblast NIH3T3 cells and murine mammary cancer C3-223 cells by retroviral infection; CSF-1R-expressing cells were isolated by FACS and two populations were obtained, named as NIH-Fms and C3-Fms. Cell surface expression was analyzed by flow cytometry. (**E**) Parental NIH3T3 fibroblast cells, NIH-Fms fibroblast cells, parental mammary C3-223 cells, and mammary C3-Fms cells were cultured in the presence of CSF-1 for 24 h and total RNA was isolated for analysis. The values are expressed as mean ± s.d. of 3 independent experiments. ns = not significant, * *p* < 0.05, ** *p* < 0.01. Data presented here are representative of 3 independent experiments.

**Figure 6 ijms-23-16028-f006:**
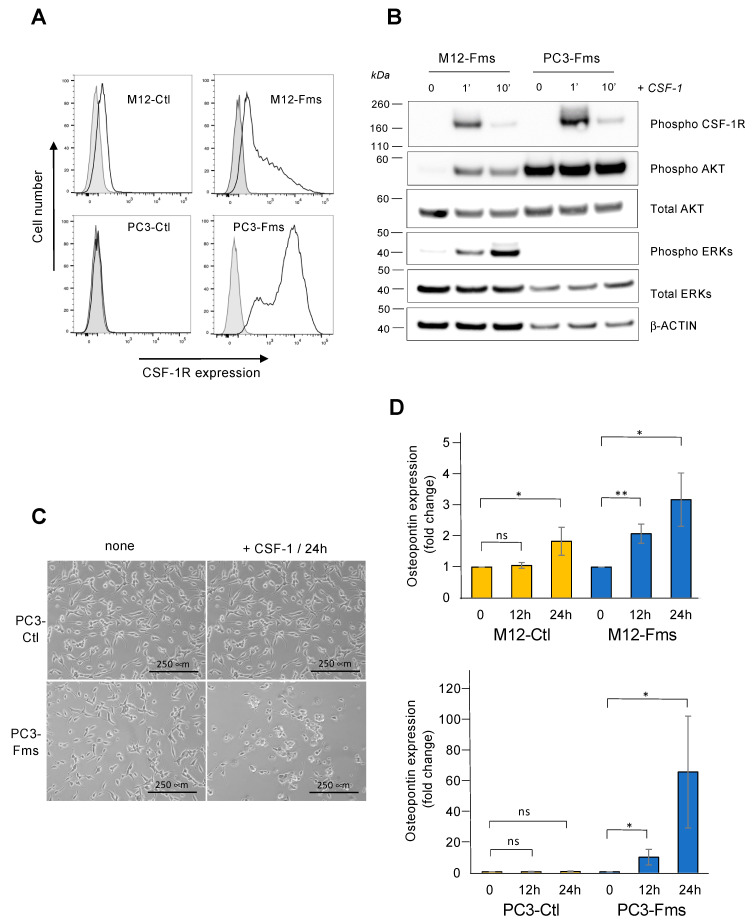
The CSF-1 receptor is functional in human prostate cells and increases spp1/osteopontin gene expression. (**A**) Two human prostate cancer cell lines, M12 and PC3, were transfected with human c-fms cDNA encoding CSF-1R or with the empty vector as a control (Ctl). Selected cell populations were analyzed for cell surface expression of CSF-1R by flow cytometry: each cell population transfected either with an empty vector (M12-Ctl and PC3-Ctl) or with c-Fms-vector (M12-Fms and PC3-Fms) were stained with anti-CD115-PE antibody (empty graph) or no antibody (grey filled graph). (**B**) Western blot analysis of CSF-1R signaling in Fms-expressing prostate cells after CSF-1 stimulation for 1 and 10 min. All cropped blots were run under the same experimental conditions. (**C**) Representative photographs (n = 3) of PC3-Ctl cells (upper panels) and PC3-Fms cells (lower panels) cultured in the absence (left panels) or the presence of CSF-1 (right panels) for 1 day. (**D**) Relative expression of spp1/osteopontin mRNA was assessed by qRT-PCR performed in triplicate (normalized against RPLP0). M12-Clt and M12-Fms cells (upper panel) and PC3-Ctl and PC3-Fms cells (lower panel) were cultured in the absence or the presence of CSF-1 for 12 h and 24 h. The values are expressed as mean ± s.d. of 3 independent experiments. * *p* < 0.05, ** *p* < 0.01, ns = not significant.

## Data Availability

The data discussed in this publication have been deposited in NCBI’s Gene Expression Omnibus [76] and are accessible through GEO Series accession number GSE216628 https://www.ncbi.nlm.nih.gov/geo/query/acc.cgi?acc=GSE216628 (accessed on 8 December 2022).

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
