# Peer review of "Macrophage-Colony-Stimulating Factor Receptor Enhances Prostate Cancer Cell Growth and Aggressiveness In Vitro and In Vivo and Increases Osteopontin Expression"

_ijms, 2022, doi:10.3390/ijms232416028_

Round 1
Reviewer 1 Report
It is acceptable with a minor revision.
Please add a new reference after this sentence: .......these processes and to identify biomarkers of metastatic CRPC [3].
Canatan H, Halifeoglu I, Caylak E. A new tumor marker in the diagnosis of prostate diseases: hK2. Klinik Laboratuvar Araştırma Dergisi, 2004; 8(2): 61-68.Author Response
Thank you for the general comment of the reviewer. Indeed, human Kallikrein2 is an interesting prostate cancer biomarker. We are deeply sorry but we were unable to have access to this reference (Canatan et al., 2004) on PubMed. We hope that the review of Rebello et al., 2021, (reference #3) will be sufficient to provide the readers with a good overview of this topic.
Reviewer 2 Report
Mougel et al. have demonstrated the functionality of CSF-1R in prostate cancer cells and have unveiled osteopontin as a CSF-1R-target gene. The manuscript is a significant contribution to the field and could be published. However, there are some minor suggestions;
1- You showed that upon CSF-1 binding, the receptor autophosphorylates and activates multiple signaling pathways in prostate tumor cells. Biological experiments demonstrated that the CSF-1R/CSF-1 axis conferred significant advantages in cell growth and cell invasion in vitro. I would like to suggest making the title of your manuscript more informative by adding information about the unveiled molecular pathways in your research.
2- The quality of some figure, especially the FACS figures, are not sufficient. Please consider improving them by using better software, such as Flowjo.
Author Response
Thank you very much for the general comment of the reviewer. Concerning the two minor suggestions:
- Concerning the signaling pathway, we have only investigated the two main signaling pathways of receptor tyrosine kinases (AKT and ERK). This is why we thought that it was not necessary to mention these pathways in the title. But we agree with the reviewer concerning the biological experiments performed both in vitro and in vivo. We thus modified the title into: “Macrophage-Colony-Stimulating Factor receptor enhances prostate cancer cell growth and aggressiveness in vitro and in vivo, and increases osteopontin expression”.
- Concerning the FACS figures, the comment was right since the FACS Figure 1C was not made with Flowjo on the contrary to the two other FACS figures (5D and 6A). We have thus performed a new flow cytometric analysis and we have used FlowJo to present these data in the new Figure 1C.
Reviewer 3 Report
Mougel et al. have investigated the functionality of the wild-type CSF-1 receptor in prostate tumor cells and identified molecular mechanisms underlying its ligand-induced activation. In this study, authors showed that upon CSF-1 binding, the receptor autophosphorylates and activates multiple signaling pathways in prostate tumor cells. The study performed by the team is classic and novel and has my high recommendation. However, I have some concerns which author should address:
1. There are minor spelling and grammar checks required. E.g., the spelling of "cleavage" is written wrong.
2. Why does the significant test show "?" besides the asterisk (Fig 5) on the graphs?
3. The authors stated that CSF-1 receptor expression promotes prostate C2H cell growth and invasion. Please provide an image showing the invasion assay. It may be using ibidi or scratch wound assay.
4. The authors claim that the CSF-1 receptor is functional in human prostate cells and increases spp1/osteopontin gene expression. Why did prostate cancer cell lines M12 and PC3, show opposite ERK phosphorylation? What could be the reason?
Author Response
Thank you very much for the general comment of the reviewer. Concerning the four concerns of the reviewer:
- We apologize for this. We checked the manuscript carefully and we have indeed corrected few grammar and spelling errors, including “cleavage”.
- We are sorry but we were unable to see “?” besides the asterisk in our Figure 5. We guess that this is a problem from our computer. If the manuscript is accepted for publication, we will pay a special attention to this point in Figure 5 during the proofreading, and we hope it could easily be corrected.
- In response to this request, we made a new Figure 2 showing representative photographs of the invasion assay: now Figure 2C with this new legend: “(C) Representative photographs (n=3) of each cell populations after 5 days of culture in the presence (+) or absence (-) of CSF-1; cells were stained with calcein AM. Scale bars, 100 µm”
- We agree with the reviewer that this result needed more explanation. Interestingly, we found a publication from the laboratory of James McCubrey that could explain our result (John T. Lee, Jr., Linda S. Steelman, William H. Chappell & James A McCubrey (2008) Akt inactivates ERK causing decreased response to chemotherapeutic drugs in advanced CaP cells, Cell Cycle, 7:5, 631-636, DOI: 10.4161/cc.7.5.5416). In this study, they showed that in prostate cancer PC3 cells, which lack PTEN, increased AKT activation led to inactivation of the MAPK/ERK pathway via Raf-1.
Accordingly, we have modified the text of our manuscript (lines 283-289): “In PC3-Fms cells, CSF-1 stimulation properly induced autophosphorylation of its receptor (Figure 6B). AKT was highly phosphorylated in the absence of CSF-1 and no detectable increase in AKT phosphorylation was observed in response to CSF-1 (Figure 6B). Although CSF-1R is functional in these cells, no ERK phosphorylation was observed in response to CSF-1 (Figure 6B). These results were similar to those demonstrating that in PC3 cells, the absence of PTEN induced constitutive AKT activity, leading to the inactivation of the MAPK/ERK pathway [35] ». We have replaced the former reference #35 (Persad et al. 2000) by this more informative reference (Lee et al., 2008).
Reviewer 4 Report
This is an excellent paper with several significant findings, especially that OPN is a target gene of CSF-1 receptor. The series of experiments to show cell signaling effects and thus functionality are appropriate to the question being asked. Future work by this team will be important.
2 comments only:
1. General: Sample size and number of experiments is missing in figures 1, 2, 4, and 5. This should be added.
2. Line 49: typo; counties should be countries
Author Response
Thank you very much for the general comment of the reviewer. Concerning the two comments of the reviewer:
- We have added the number of experiments in the text of the legends of the different figures, and the sample size for Figure 3 ( "For all these experiments, each group contained eight mice").
- We apologize for this. This typo, and few others, have been corrected